# Retrieval-Augmented Reasoning for Extremely Low-Resource Language Decipherment

## Abstract

Inspired by linguistic Olympiads, extremely low-resource language reasoning presents a unique challenge that enables models to solve problems without prior knowledge. This task mirrors the Rosetta Stone decipherment process, where the goal is to induce and apply linguistic rules from minimal context. Existing methods mainly rely on naive in-context learning that fails to handle the complexity and diversity of language rules. To mitigate this issue, we propose a framework that combines dynamic knowledge construction with task-aware retrieval augmentation. First, we use large language models (LLMs) to generate a diverse set of task-specific examples that instantiate potential linguistic rules for the target low-resource language. Second, we apply a semantic retrieval mechanism to select the most relevant examples for each test query, preventing context overload and ensuring focused, analogical reasoning. Our method shifts from learning language distributions to dynamically discovering and applying rules. Experimental results on the LINGOLY and Linguini benchmark show that our approach achieves competitive performance across various LLMs, outperforming existing baselines. More importantly, our framework advances extremely low-resource reasoning and provides a generalizable framework for rule induction under knowledge constraints.

## 1 Introduction

Deciphering unknown linguistic systems is a hallmark of human intelligence, exemplified by the Rosetta Stone (Bozhanov & Derzhanski, 2013). The Rosetta Stone question presents paired examples of a low-resource language alongside its English translation, requiring solvers to deduce the language's grammar and vocabulary solely from those minimal clues. In particular, the Rosetta Stone is incomplete, thus the problem poses the challenge that there is no explicit knowledge to help solve it. For machine learning models, to do this well, it must possess multiple capabilities, including summarizing grammatical, morphological and semantic rules from several examples and applying them to new problems. This task is significant for advancing interpretability, reasoning, and applications in historical linguistics and endangered language documentation. Moreover, it provides a computational testbed for analogical reasoning and systematic generalization.

As a conversion to the deciphering problem of the Rosetta Stone, recent benchmarks like LINGOLY (Bean et al., 2024) and Linguini (Sánchez et al., 2024) formalize this challenge for machine learning models, presenting tasks in low-resource or artificial languages, ensuring models cannot rely on memorized knowledge. These datasets present a formidable challenge for current large langue models (LLM) with powerful reasoning abilities. As shown in Figure 1, each problem is designed as a Rosetta-style task, where the model needs to infer linguistic rules solely from the provided context or require deductive reasoning, such as translating unseen sentences, aligning word pairs, or inferring morphological rules. Since the problems come from extremely low-resource or extinct languages, the LLMs do not have knowledge of those languages and

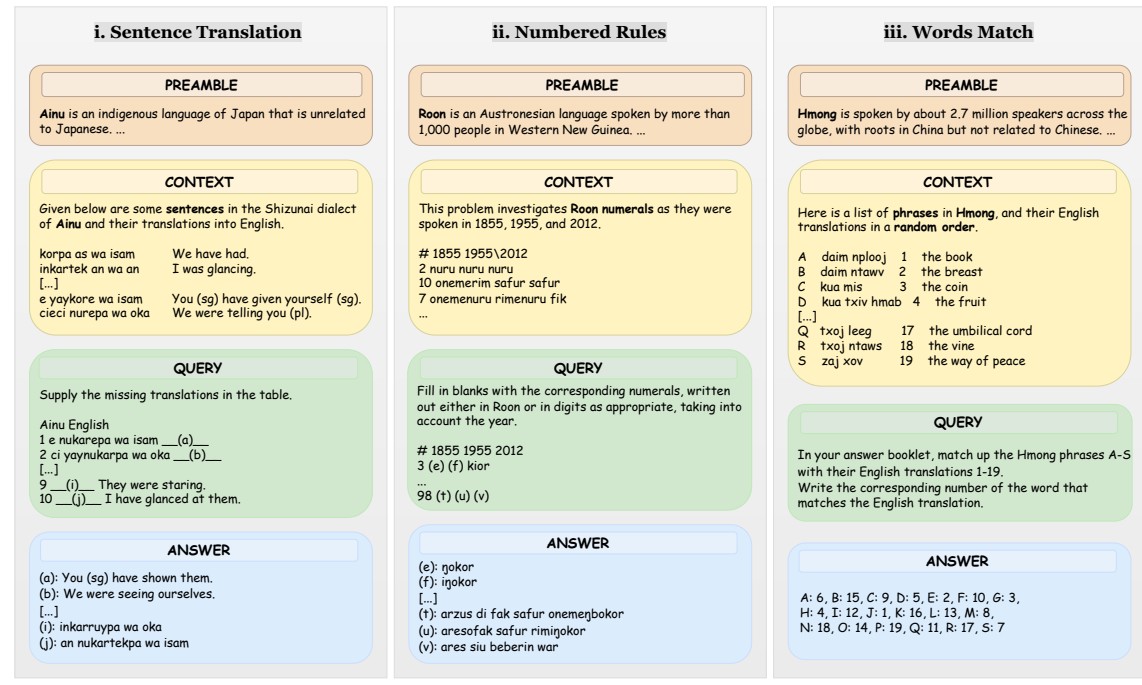

Figure 1: Rosetta Stone linguistic problem example. Each question contains four parts, PREAMBLE: introducing the background information of the current low resource language, CONTEXT: a small number of translation pairs between the current language and English, QUERY: the question set based on context, and ANSWER: the correct answer corresponding to query.

can only attempt reasoning and deduction without prior knowledge of the target language. As a result, the central scientific problem is: ***How can models induce and apply linguistic rules from minimal contextual examples in unfamiliar languages?***

Recent work on deciphering low-resource languages with LLMs has relied on several strategies of increasing sophistication. On one hand, researchers have explored few-shot and chain-of-thought prompting, where models are asked to generalize from a handful of translation exemplars to uncover grammatical rules in unseen languages. However, this approach only captures surface patterns, which fails to capture deeper cross-linguistic patterns. These methods generally struggle when there is no training sample for the target language. On the other hand, the advanced approach is analogical prompting (Ramji & Ramji, 2025), where auxiliary exemplars are automatically generated in related, higher-resource languages and combined with target examples. This two-stage reasoning procedure enables models such as GPT-4o and Llama-3.1-405B to leverage their latent multilingual knowledge more effectively. However, these studies reveal the limitations of current methods: Although analogical and chain-of-thought prompting improve over naive few-shot learning, models still struggle with multistep reasoning, complex morphosyntactic generalizations, and robust handling of languages absent from pretraining. This gap underscores the need for novel approaches to reasoning-based decipherment in low-resource settings.

To alleviate this gap, we introduce a framework through two stages: **Dynamic Knowledge Construction** and **Task-Aware Retrieval Augmentation**, which realizes the decipherment reasoning with dynamic and flexible linguistic rules instead of factual data. In particular, for the **Dynamic Knowledge Construction**, we leverage LLMs to generate diverse rule-guided exemplars, forming a dynamic knowledge base. The

first stage, Dynamic Knowledge Construction, is necessary to proactively fill the fundamental knowledge vacuum by generating a rich set of rule-embodying examples, moving beyond the limited patterns present in the original context. However, the sheer volume of generated knowledge necessitates the second stage, Task-Aware Retrieval Augmentation, which acts as a critical filter to mitigate information overload and ensure that the model's reasoning is guided by the most pertinent analogies for each specific problem.

For the **Task-Aware Retrieval Augmentation**, to reduce inefficiency and noise, we retrieve the top-$K$ most relevant examples for each test query, enabling focused analogical reasoning. In particular, the necessity of retrieval in our framework is grounded in the two theoretical foundations: **(1) Cognitive Load Theory.** The limited context window of transformers mirrors human working memory (Leng et al., 2024). By retrieving only the top-K most relevant examples, we reduce extraneous cognitive load, enabling the model to focus on the key rule patterns necessary for solving the task. **(2) Analogical Reasoning Theory.** Human problem-solving often relies on analogy, transferring knowledge from similar past cases (Wang et al., 2024; Musker et al., 2024; Wei et al., 2022). Our BM25 retriever functions as a mechanism for identifying analogous cases (e.g. "What was the correct form for an adjective describing a 'house' in a similar sentence?"), allowing the model to apply analogical reasoning effectively. Beyond benchmark performance, the broader implications of this work are profound. Success in low-resource language inference contributes to the revitalization of endangered languages, improves multilingual adaptability in real-world settings, and offers a computational account of how humans induce rules from limited evidence. Overall, our contributions are both methodological and conceptual:

- We formalize low-resource language decipherment as a dynamic rule induction task and achieves competitive performance on the LINGOLY and Linguini, compared with existing methods.
- Our proposed framework combines rule generation and retrieval to simulate human-like decipherment, which is applicable to different types of Rosetta Stone problems, such as sentence translation and words match.
- Experimental results demonstrate that the approach is effective based on both open-source and commercial LLMs with various sizes and can further stimulate LLMs with strong inherent reasoning capabilities.

## 2 RELATED WORK

The approaches that can be transferred to solve the decryption problem of the Rosetta Stone are to infer linguistic rules and conduct deductive reasoning. The first branch of the approaches is *in-context learning* (ICL), where the ability of LLMs to learn from examples provided in the prompt (Brown et al., 2020; Li, 2023; Luo et al., 2024). Standard approaches to our task, such as directly providing context examples, rely on this capability. However, the performance of ICL is often limited by the number of examples available in the context (typically 10). In contrast, our work enhances ICL by massively scaling up the number of potential examples through generation and then intelligently selecting the most relevant ones. This augmentation goes beyond the constraints of the original problem's limited context, allowing for more efficient and focused reasoning. The second branch of the approaches is the *Retrieval-Augmented Generation* (RAG) (Lewis et al., 2020; Sánchez et al., 2024; Zhao et al., 2024; Zhang et al., 2024; Fan et al., 2025) which integrates parametric knowledge stored in model weights with non-parametric knowledge from external corpora via a retriever. Based on the paradigm of RAG, we transfer this methodology into alleviating the issue of the Rosetta Stone decryption.

Moreover, in comparison to the popular RAG methods, the knowledge database needs to be constructed from-scratch in our scenarios. LLMs have been used to generate synthetic data for training or enhancement (Anil et al., 2023), while our approach does not generate data to modify the model weights (e.g., by fine-tuning). Instead, we generate contextual knowledge on-the-fly, which serves as non-parametric, in-context clues (Wang et al., 2023). This is akin to "self-generated prompts" or "knowledge distillation" from the model, focused on instantiating linguistic rules inferred directly from the problem context. Our approach

enables flexible, real-time generation of task-specific knowledge without the need for fine-tuning (Zhou et al., 2022), distinguishing it from typical synthetic data generation approaches.

There are several other reasoning tasks related to decryption which requires rule induction. **Cryptography and Puzzle Solving.** Deciphering an unknown language is akin to breaking a cipher (e.g., substitution ciphers) (Jakobsen, 1995), where the model must find mappings and patterns between the unknown language and a known one. **Inductive Logic Programming.** Inductive Logic Programming (ILP) (Muggleton, 1991; Mooney, 1996; Raedt, 2008) aims to learn logical rules from positive and negative examples, typically requiring formalizing features into logical predicates, a process that is challenging for complex, low-resource linguistic phenomena. **Meta-Learning and Multi-Task Learning.** Training models on a distribution of related tasks (e.g., many language inference problems) to acquire a "learning-to-learn" capability is a promising direction (Hospedales et al., 2022; Gharoun et al., 2024). However, this approach requires large, diverse training data and risks overfitting to the seen task distribution. Our method, by contrast, is a zero-shot, in-context approach that requires no additional training, making it more generalizable to entirely novel languages and tasks.

In summary, prior efforts have either emphasized parametric learning (ICL, meta-learning) or static augmentation (synthetic data generation). However, these approaches struggle when the target distribution is entirely unseen and cannot be inferred from memorized knowledge. Our method differs fundamentally in that it treats the problem as an online reasoning task: knowledge is not pre-stored but dynamically constructed and filtered. This distinction allows our framework to better simulate the human decipherment process, where knowledge is actively hypothesized and revised in real time rather than recalled from a fixed memory. In this sense, our work bridges the gap between information retrieval, analogical reasoning, and computational linguistics.

## 3 METHOD

### 3.1 OVERVIEW OF RETRIEVAL-AUGMENTED SYNTHETIC EXEMPLAR GENERATION

We denote the low-resource language reasoning dataset as $D = (x_1, y_1), ..., (x_n, y_n)$, where each instance $x_i$ consists of

- **context** $C_i$: a short passage containing a few surface forms (sentences, phrases or numerals) in an unseen language $L_i$
- **query** $Q_i$: a question about the underlying rule of $L_i$.
- **gold answer** $y_i$.

The model involves $C_i$ and $Q_i$ at test stage and predicts $y_i$ without any prior knowledge of $L_i$. Our goal is to lift the model's performance on $y_i$ by automatically enriching its working memory with synthetic, task-relevant exemplars of $L_i$.

To alleviate this problem, we propose RASEG (Retrieval-Augmented Synthetic Exemplar Generation), a two-stage pipeline, as illustrated in 2. The framework consists of two components:

- **Offline Generation (§3.1.1).** Deepseek-R1 writes synthetic exemplars for every language L in D at three linguistic levels (sentence, phrase, number). These exemplars are stored in a retrieval index R.
- **Online Inference (§3.1.2).** Given a test instance (C, Q) in language L and level $\ell$, the system retrieves the top-k most semantically similar exemplars from R and concatenates them to C before feeding the enriched prompt to the reasoning model.

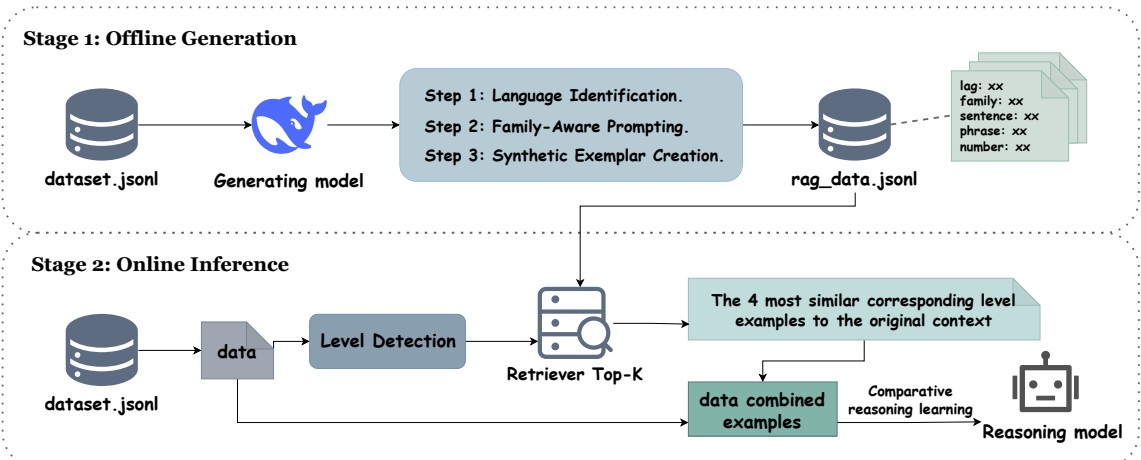

Figure 2: Overview of RASEG. During the Offline Generation stage, we identify the low-resource language L and generate 15 synthetic exemplars per level. During the Online Inference stage, for each test example we retrieve the top-k exemplars and prepend them to the prompt.

### 3.1.1 OFFLINE GENERATION OF R

We populate the retrieval index R in three steps.

**Step 1: Language Identification.**
For every example $(x, y) \in D$, we utilize DeepSeek-R1 with a zero-shot prompt: "Identify the low-resource language in the following context. Return only the ISO 639-3 code.", to obtain L.

**Step 2: Family-Aware Prompting.**
We also query DeepSeek-R1 for the language family F(L). This meta-information is later used to bias exemplar generation towards typologically similar languages, increasing the chance of capturing relevant morphosyntactic patterns.

**Step 3: Synthetic Exemplar Creation.**
For each pair $(L, F(L))$ we prompt DeepSeek-R1 to produce 5 × 3 = 15 synthetic exemplars: 5 sentence-level ⟨low-resource, English⟩ translation pairs, 5 phrase-level pairs and 5 number-level pairs.

Since the languages of each data in the dataset are repeated, the total number of synthetic exemplars in some languages exceeds 15. All exemplars are assigned a key-value record and appended to R. The prompt used to generate additional examples is in Appendix A.

### 3.1.2 ONLINE INFERENCE

Given a test instance (C, Q) we perform retrieval as follows.

**Step 1: Level Detection.**
We classify the task granularity $\ell$ with a rule-based trigger set:
$tokens \in \{"phrases", "words", "phrase", "word"\} \longrightarrow \ell = phrase$
$tokens \in \{"numbers", "numerical", "number"\} \longrightarrow \ell = number$
$otherwise \longrightarrow \ell = sentence.$

**Step 2: Retrieval.**
We query R with $lag = L$ and $level = \ell$, then rank the exemplars by the similarity of the retriever to C. The top-k exemplars $E_1 \ldots E_k$ are concatenated to C in order of similarity.

**Step 3: Reasoning.**
Put the retrieved enhanced test cases (C, q) into the designed prompt and input them to model for reasoning. The prompts used to ultimately enable the model to reason are in Appendix C.

## 3.2 DYNAMIC KNOWLEDGE CONSTRUCTION

To construct the knowledge database from-scratch for decipherment, we adopt a dynamic knowledge construction method with several key aspects. A crucial challenge in this stage is ensuring the quality of generated exemplars. Simply producing thousands of sentences risks introducing noise, contradictions, or degenerate cases. To mitigate this, we employ a two-step quality control pipeline. First, we apply automatic filtering based on lexical diversity and structural validity, ensuring that generated examples are not trivial restatements of the original context. Previous augmentation relies on prior distributional knowledge, while our approach is entirely self-contained, using only the current instance's preamble and context. Second, we conduct rule consistency checks by prompting the LLM to verify whether new exemplars adhere to the inferred morphological or syntactic patterns. This iterative self-verification process significantly reduces spurious examples and improves retrieval efficiency downstream. Moreover, by stratifying examples across task levels (e.g., word-level vs sentence-level reasoning), we create a balanced knowledge base that better matches the granularity of incoming queries. Traditional augmentation seeks to increase diversity within a known distribution. Our method generates a *de-novo* knowledge source to compensate for the absence of parametric knowledge in low-resource languages.

## 3.3 RETRIEVAL AUGMENTATION

Given the large number of generated examples, retrieval is essential to prevent information overload. Key considerations include:

- **Information Overload.** The transformer's context window is limited, and including too many examples would push out critical information, like the preamble and question.
- **Semantic Relevance.** Relevant examples must be semantically aligned with the query. For instance, examples involving "house" and "green" are more useful for a translation task than those involving "apple" or "dog."
- **Avoiding Conflicting Rules.** Since multiple rules may exist, retrieval ensures that the most relevant examples are selected, reducing the risk of rule conflicts.
- **Simulating Human Focus.** Similar to how a linguist recalls analogous cases, retrieval enables the model to focus on the most relevant examples.

Although dense retrievers such as BGE or Qwen3-Embedding prioritize semantic similarity, the term frequency scoring via sparse retrieval ensures that retrieved examples exhibit exact lexical overlap, which is crucial for linguistic rule extraction in low-resource settings. This explicit overlap provides directly comparable exemplars, helping the model focus on fine-grained rules.

## 4 EXPERIMENTS

### 4.1 SETUP

We conducted experiments on two Olympiad-level linguistic reasoning datasets, including LINGOLY (Bean et al., 2024) and Linguini (Sánchez et al., 2024). We utilize EM, Chrf (Popovic, 2015), BLEU (Papineni et al., 2002) and ROUGE (Lin, 2004) as the evaluation metrics, following the settings of LINGOLY and Linguini. We use Deepseek-R1 for generation and evaluate reasoning on QwQ-32B, DeepSeek-R1-Distill-Qwen-32B (Referred to as Qwen-32B in the following text), Qwen-3-32B, Deepseek-R1, and GPT-4o-mini. Both the sparse (BM25) and dense retriever (BGE) are selected at the online stage.

### 4.2 BASELINES

We compare our method with three types of baselines: (1) **No context.** Directly querying the LLM without any context. (2) **Only context.** Providing the original preamble and context from the problem. (3) **Inductive Linguistic Reasoning (Ramji & Ramji, 2025).** Generating examples from a related high-resource language and adding them to the context, is abbreviated as Inductive. Furthermore, the no context method is one of the original baselines in the LINGOLY benchmark. For the Linguini dataset, the no context method is not used because the question contains the reference context naturally (e.g., the context is a disordered English and low resource language phrase, and the question is the corresponding order of answering these short words).

Table 1: The results of various methods on the LINGOLY dataset. The best scores are **bold**.

| Model | QwQ-32B | | Qwen-32B | | Qwen-3-32B | | Deepseek-R1 | | GPT-4o-mini | |
|---|---|---|---|---|---|---|---|---|---|---|
| Metric(↑) | EM | Chrf | EM | Chrf | EM | Chrf | EM | Chrf | EM | Chrf |
| No context | 13.85 | 25.65 | 14.69 | 27.15 | 16.34 | 29.40 | 19.92 | 29.74 | 17.96 | 31.87 |
| Only context | 27.06 | 40.95 | 22.65 | 45.42 | 30.92 | 51.12 | 38.92 | 46.48 | 27.01 | 48.23 |
| Inductive | 30.64 | 47.35 | 24.77 | 46.11 | 33.27 | 53.47 | 42.99 | 52.13 | 28.57 | 49.82 |
| RASEG | 34.04 | 55.58 | 23.23 | 44.27 | 33.53 | 54.62 | 42.39 | 54.57 | 26.06 | 47.57 |
| RASEG + Inductive | **36.47** | **57.91** | **24.82** | **46.31** | **33.85** | **55.49** | **45.55** | **55.04** | **28.76** | **50.42** |

Table 2: The results of various methods on the Linguini dataset. The best scores are **bold**.

| Model | QwQ-32B | | Qwen-32B | | Qwen-3-32B | | Deepseek-R1 | | GPT-4o-mini | |
|---|---|---|---|---|---|---|---|---|---|---|
| Metric(↑) | EM | Chrf | EM | Chrf | EM | Chrf | EM | Chrf | EM | Chrf |
| Only context | 8.79 | 27.83 | 3.65 | 29.14 | 7.67 | 31.68 | 15.70 | 46.15 | 5.34 | 26.64 |
| Inductive | 8.81 | **35.91** | 5.41 | 27.51 | 8.93 | **33.25** | 15.72 | 45.18 | 5.27 | 29.29 |
| RASEG | **11.79** | 35.10 | 6.63 | 28.75 | 9.99 | 32.23 | **19.83** | 47.24 | **7.72** | **30.27** |
| RASEG + Inductive | 11.16 | 35.62 | **6.68** | **30.32** | **10.80** | 33.13 | 18.00 | **48.67** | 6.48 | 28.76 |

### 4.3 MAIN RESULTS

As shown in Table 1 and 2, our proposed method reveals a substantial advantage. In LINGOLY, our method achieves the highest reported performance across all models. For instance, Deepseek-R1 achieves 45.55 EM and 55.04 Chrf, surpassing the baseline by more than 3 points in EM and nearly 3 point in Chrf. In particular, even stronger relative gains are observed for open-weight models, where QwQ-32B jumps from 34.04 to 36.47 EM, and Chrf climbs from 55.58 to 57.91, demonstrating that our pipeline particularly benefits models with less built-in linguistic knowledge. Conversely, performance on Linguini is more compressed,

yet the same pattern holds. Based on the Deepseek-R1, our method peaks at 19.83 EM and 47.24 Chrf, and remains stable at 18.00 EM and 48.67 Chrf when combined with inductive examples, while all other models see modest but consistent improvements. Notably, GPT-4o-mini underperforms relative to its size on both benchmarks, suggesting that scale alone is insufficient without targeted rule-based augmentation. Taken together, the quantitative and qualitative evidence confirms that dynamically generating and retrieving rule-centric exemplars is a robust strategy for low-resource language inference, pushing model performance well beyond the ceiling imposed by static or purely parametric approaches.

Table 3: Comparison of BM25 and BGE Retriever in terms of EM scores with various types of problems based on different LLMs.

| QwQ-32B | BM25 | BGE |
|---|---|---|
| Compounding | 34.92% | 30.16% |
| Morphology | 30.07% | 30.39% |
| Numbers | 18.95% | 18.95% |
| Phonology | 33.33% | 30.11% |
| Semantics | 25.37% | 32.09% |
| Syntax | 34.44% | 33.89% |

| Qwen-32B | BM25 | BGE |
|---|---|---|
| Compounding | 28.57% | 23.81% |
| Morphology | 19.28% | 15.36% |
| Numbers | 3.16% | 5.26% |
| Phonology | 26.90% | 29.24% |
| Semantics | 20.15% | 18.66% |
| Syntax | 35.00% | 28.89% |

| GPT-4o-mini | BM25 | BGE |
|---|---|---|
| Compounding | 19.04% | 20.63% |
| Morphology | 18.95% | 20.91% |
| Numbers | 5.26% | 5.26% |
| Phonology | 30.99% | 30.12% |
| Semantics | 26.12% | 26.12% |
| Syntax | 37.22% | 37.78% |

| Deepseek-R1 | BM25 | BGE |
|---|---|---|
| Compounding | 26.98% | 23.57% |
| Morphology | 41.17% | 40.42% |
| Numbers | 29.47% | 25.52% |
| Phonology | 39.47% | 39.27% |
| Semantics | 33.58% | 32.07% |
| Syntax | 50.55% | 50.67% |

## 4.4 EFFECTS OF RETRIEVERS

As shown in Table 3, we investigate the effectiveness of our framework with different retrievers. The results demonstrate that in most cases, the sparse retrieval engine BM25 is superior to the dense retrieval engine (e.g., BGE), which does not conform to the common trend of retrieval-augmented methods. This is due to the different task scenarios. For the problem of decipherment, the better retrieval mechanism leans towards exact matching. For instance, if the original context contains the word "travel", bm25 tends to search for examples with the word "travel", while BGE and other dense retrievers tend to search for those with high semantic similarity (such as "trip" first). Our task happens to be to find out what the low-resource language corresponding to "travel" is. Therefore, we need examples that have even more high degree of overlap with the original context, to facilitate model comparison and learning.

Table 4: The results obtained by taking different K values when using the BM25. The best scores are **bold**.

| Dataset | LINGOLY | | | | Linguini | | | |
|---|---|---|---|---|---|---|---|---|
| Model | QwQ-32B | | Deepseek-R1 | | QwQ-32B | | Deepseek-R1 | |
| Metric(↑) | EM | CHRF | EM | CHRF | EM | CHRF | EM | CHRF |
| k=2 | 32.13 | 50.22 | 41.93 | 51.98 | 9.78 | 34.78 | 18.61 | 45.13 |
| k=3 | 30.36 | 49.84 | 41.01 | 53.27 | 9.87 | **35.89** | 18.48 | 47.84 |
| k=4 | **34.04** | **55.58** | **42.39** | **54.57** | **11.79** | 35.10 | **19.83** | 47.24 |
| k=5 | 30.61 | 48.52 | 40.01 | 51.54 | 9.74 | 35.52 | 19.57 | **47.90** |

Furthermore, as shown in Table 4, we analyze the performance of the hyper-parameter of top-k for the retrievers. The results show that the model achieves better performance when k=4. When k is too large, redundant information interferes with the model's deductive reasoning, while when k is too small, the amount of information provided is insufficient, making it difficult to offer clues for reasoning.

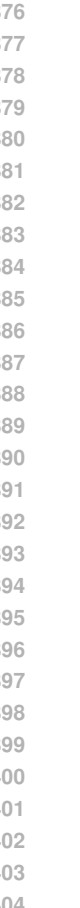
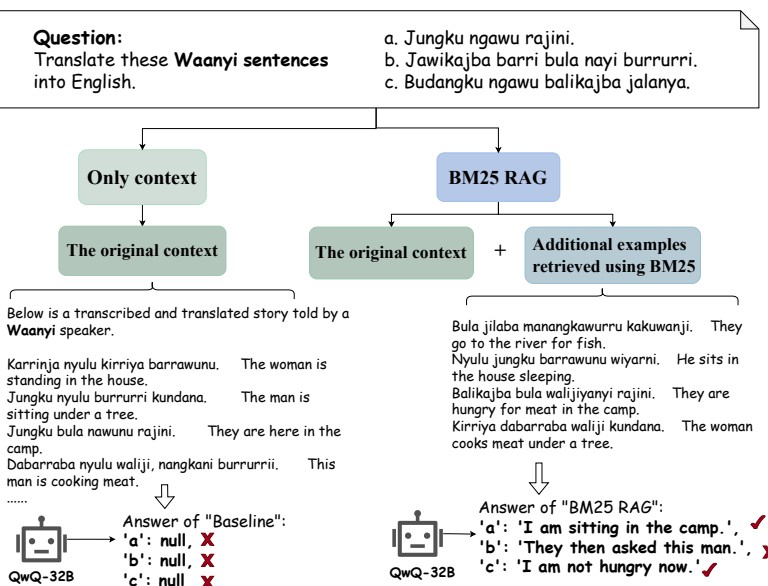

Figure 3: Examples of the only context and our framework on the LINGOLY in the sentence translation.

## 4.5 CASE STUDY

To further illustrate the efficacy of our framework, we present two representative cases from the LINGOLY and Linguini datasets, highlighting how dynamic knowledge construction and task-aware retrieval enable accurate reasoning in truly unfamiliar linguistic systems. As shown in Figure 3, the model is tasked with translating Waanyi sentences such as "Jungku ngawu rajini" into English. The original context provides only sparse sentence-level translations, and the baseline method fails to infer any correct output. Our method, however, retrieves synthetic exemplars that explicitly encode Waanyi's ergative-absolutive alignment and spatial deixis, such as "Jungku bula nawunu rajini" mapped to "They are here in the camp." This retrieved example not only clarifies the lexical meaning of "rajini" as "in the camp," but also reinforces the subject-verb-object pattern absent in the original context. Consequently, the model correctly translates "Jungku ngawu rajini" as "I am sitting in the camp," demonstrating how analogical reasoning over retrieved rule-instantiating examples can compensate for initial knowledge scarcity. Moreover, we conduct a case analysis for Linguini datasets. More details are provided in Appendix E.

## 5 CONCLUSION

We propose a framework for low-resource language inference that combines dynamic knowledge generation with retrieval augmentation. By shifting from memorizing distributions to dynamically inducing rules, our approach simulates human decipherment and addresses two core challenges in low-resource language decipherment: **(i) Knowledge Scarcity.** The dynamic knowledge construction phase generates the necessary "raw material" (rule examples) for reasoning. **(ii) Focused Reasoning.** The retrieval phase directs the model's limited attention to the most relevant examples, enabling efficient analogical reasoning and reducing cognitive load. Thus, our method translates human-like linguistic decipherment into a computationally scalable framework.

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

## A PROMPT USED FOR KNOWLEDGE GENERATION.

The following is the prompt used to generate additional examples of each data in LINGOLY and Linguini datasets using the Deepseek-R1 model. The prompt word first introduces the composition of each data and the need to generate knowledge examples. The processing flow of deepseek-r1 model is to first identify the low resource language name corresponding to the current data, and then generate more reference learning examples according to the current context examples, including the examples of "sentence", "phrase" and "number", and the final return format is JSON.

```
1  prompt=f'''
2      Now I have a low resource language reasoning task, and the specific task
       content is to answer questions in each piece of data, including context,
       query, answer, and task_type.
3
4      However, there is currently a problem that there are too few examples of
       context provided in the dataset, making it difficult for the model to
       answer questions correctly based on these examples. Therefore, I would like
        to generate some examples that the model can refer to and create a
       universal retrieval library. The specific method is to first read the
       context of each piece of data. You need to identify what low resource
       language is in the current data and determine the language family to which
       this language belongs. Then, select high resource languages in the same
       language family that have similar language features to the current language
       . Next, please generate examples of translation pairs between these high
       resource languages and English. I will put the translation pairs you
       generate into the retrieval library for subsequent model retrieval and
       learning.
5
6      Please note that the selected high resource language translation examples
       and the new current low resource language translation examples need to help
        the model learn the knowledge/rules of the low resource language
       corresponding to the current data, and generate three levels of translation
        pairs (sentence, phrase, and number), each level requiring five
       translation pairs. For the translation examples of high resource languages,
        I hope they are consistent with the original context examples in the data,
        so that the model can learn more directly how similar languages express
       the same phrase, and thus learn language knowledge. For example, in the
       following data, if you recognize that the current low resource language is
       language B, which belongs to language family C, and the most suitable high
       resource language is language A, then you need to generate the following
       JSON format for my search knowledge:
7
8      {
9      "lag": "Current high resource language A",
10     "family": "Language A's language family",
11     "sentence": ["Sentence 1 in A language\t Corresponding English translation
       sentence 1", "Sentence 2 in A language\t Corresponding English translation
       sentence 2", "Sentence 3 in A language\t Corresponding English translation
       sentence 3", "Sentence 4 in A language\t Corresponding English translation
       sentence 4", "Sentence 5 in A language\t Corresponding English translation
       sentence 5"],
12     "phrase": ["Phrase 1 in A language\t Corresponding English translation
       phrase 1", "Phrase 2 in A language\t Corresponding English translation
       phrase 2", "Phrase 3 in A language\t Corresponding English translation
       phrase 3", "Phrase 4 in A language\t Corresponding English translation
       phrase 4", "Phrase 5 in A language\t Corresponding English translation
       phrase 5"],
13     "number": ["Number 1 in language\t Corresponding Arabic number 1", "Number
       5 in language\t Corresponding Arabic number 5", "Number 9 in language\t
       Corresponding Arabic number 9", "Number 14 in language\t Corresponding
       Arabic number 14", "Number 27 in language\t Corresponding Arabic number
       27"]
14     }
```

```
15
16      Now please process the following data in sequence and provide the
        corresponding generated JSON results.
17  '''
```

## B    EXAMPLES OF DATA IN THE DEEPSEEK GENERATED RETRIEVAL LIBRARY.

The following is the knowledge example of the JSON format of Beja language generated by the Deepseek-R1 model.

```
18  json_data=f'''{
19      "lag": "Beja",
20      "family": "Afro-Asiatic",
21      "sentence":
22      ["Dib winu diwini.  The big wolf is sleeping.", "Ti'ari tamtiniit kitte.
         She cannot eat food.", "Uufaar ooyoo rhaabu.    The man has seen the
        flower.", "Tihatay kitdibil.  She is not collecting the horse.", "Uugwib
        kiidwiini.  The mouse is not sleeping."],
23      "phrase": ["dib winu   the big wolf", "ti'ari tamtiniit   eating food (
        feminine)", "oofaar rhaabu   seen the flower", "tihatay dibil
        collecting horse (feminine)", "uugwib diwini    sleeping mouse"],
24      "number": ["gaal   1", "ay    5", "sagal    9", "tamanyo ushu   14", "
        tamanyo malo   27"]
25  }'''
```

## C    PROMPTS USED FOR THE FINAL AUGMENTED REASONING.

The following is the prompt format used by various models for reasoning on LINGOLY datasets.

```
26  prompt=f'''Below is a problem sheet from a linguistics exam. You will first see
         the entire sheet, then be asked to respond to specific questions from the
        sheet. Your answers to the questions should rely only on reasoning about
        the information provided in the sheet.
27
28      {preamble}
29
30      {context}
31
32      === Additional {level} examples from {language} ===
33      {additional examples}
34
35      {questions}
36
37      Now respond to the following questions:
38
39      {subquestions}
40
41      Format your response as a json file with the keys as provided below:
42      {\"A\": \"\", \"B\": \"\", \"C\": \"\"}
43  '''
```

The following is the prompt format used by various models for reasoning on Linguini datasets.

```
44  prompt=f'''You are a professional linguist who is good at learning and
45      understanding low-resource languages. Please use your knowledge of
46      linguistics and semiotics (such as pronoun mapping, tense marking, number
47      base representation, etc.) to learn and understand low-resource languages
48      in context, and answer the specified questions based on the context.

        The answers you generate should not include reasoning or thinking processes
        , but directly answer questions based on the query, and the final answer
        should start with \"Final Answer:\" The following are examples of correct
        answer formats for different task types:
        Example of correct answer format for translation task: Final Answer:['I
        want a cat.','You are cute.','Do you want some water?']
        Example of correct answer format for fill_blanks task: Final Answer:['dog
        ','apple','the sun']
        Example of correct answer format for text_to_num task: Final Answer
        :['920','16']
        Example of correct answer format for num_to_text task: Final Answer:['
        eleven','one thousand']
        Example of correct answer format for match_letters task: Final Answer:['A
        ','D','F','B','E','C']

        ##Task Type:##
        {task_type}

        ##Context:##
        {context}

        ##Query:##
        {query}
    '''
```

## D  FULL DETAILED RESULTS OF LINGOLY DATASET.

Table 5 shows BLEU and ROUGE scores of the results obtained by reasoning with various methods for various models on LINGOLY dataset.

Table 5: The BLEU and ROUGE results of various methods on the LINGOLY dataset. The best scores are **bold**.

| Model | QwQ-32B | | Qwen-32B | | Qwen-3-32B | | Deepseek-R1 | | GPT-4o-mini | |
|---|---|---|---|---|---|---|---|---|---|---|
| Metric(↑) | BLEU | ROUGE | BLEU | ROUGE | BLEU | ROUGE | BLEU | ROUGE | BLEU | ROUGE |
| No context | 15.82 | 22.64 | 16.85 | 23.94 | 18.47 | 26.71 | 21.52 | 28.03 | 20.33 | 28.92 |
| Only context | 30.44 | 38.66 | 26.68 | 40.26 | 35.49 | 46.41 | 41.22 | 45.66 | 30.40 | 44.13 |
| Inductive | 34.46 | 43.66 | 27.98 | 41.93 | 38.09 | 49.63 | 45.74 | 49.85 | 32.33 | 45.97 |
| RASEG | 38.53 | 50.43 | 27.37 | 39.04 | 37.37 | 50.27 | 45.76 | 52.16 | 29.60 | 43.95 |
| RASEG + Inductive | **41.46** | **53.68** | **28.11** | **42.06** | **38.88** | **51.54** | **48.72** | **53.23** | **32.49** | **46.37** |

Figure 4, 5, 6, 7, 8 respectively show the EM scores of various methods on different question types and difficulty levels of different models. The size of the circle represents the proportion of this type of question on the entire dataset, and the percentage number on the circle represents the proportion of completely correct reasoning on this type of question.

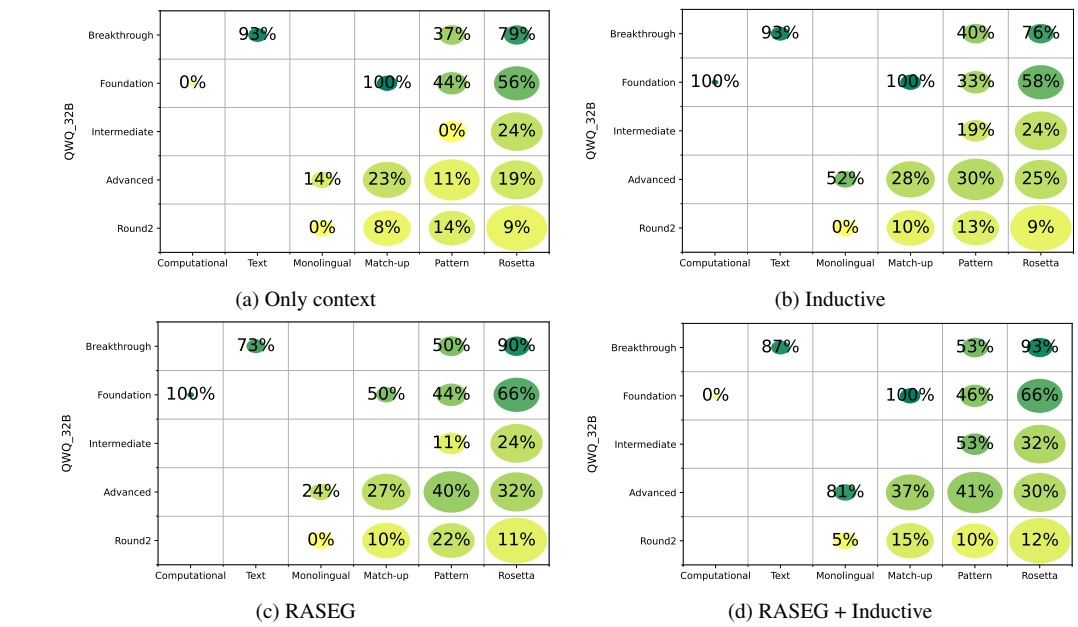

Figure 4: EM scores of various methods on different question types and difficulty levels of QwQ-32B.

# E ADDITIONAL CASE STUDY

As shown in Figure 9, we investigate the comparison between the baseline and our framework on Linguini datasets. The case 1 involves translating English clauses into Basque, such as "You (sg) touched me" and "They approached me." The challenge lies in inferring Basque's complex auxiliary selection and agreement morphology. The baseline again yields null outputs, indicating a complete failure to induce the required morphosyntactic rules. Our method retrieves synthetic exemplars like "Zuk ogia jan duzu" ("You (sg) have eaten the bread"), which instantiate the auxiliary "duzu" for second-person singular transitive verbs. Similarly, "hurbildu zaizkit" from the exemplar set directly models the intransitive auxiliary "zaizkit" used for third-person plural agents. These retrieved forms enable the model to generalize correctly, producing "ukitu nauzu" and "hurbildu zaizkit" for the target sentences. These cases underscore that our framework does not merely retrieve similar strings but rather surfaces linguistically informative exemplars that embody abstract rules, enabling systematic generalization even in morphologically rich and typologically distant languages.

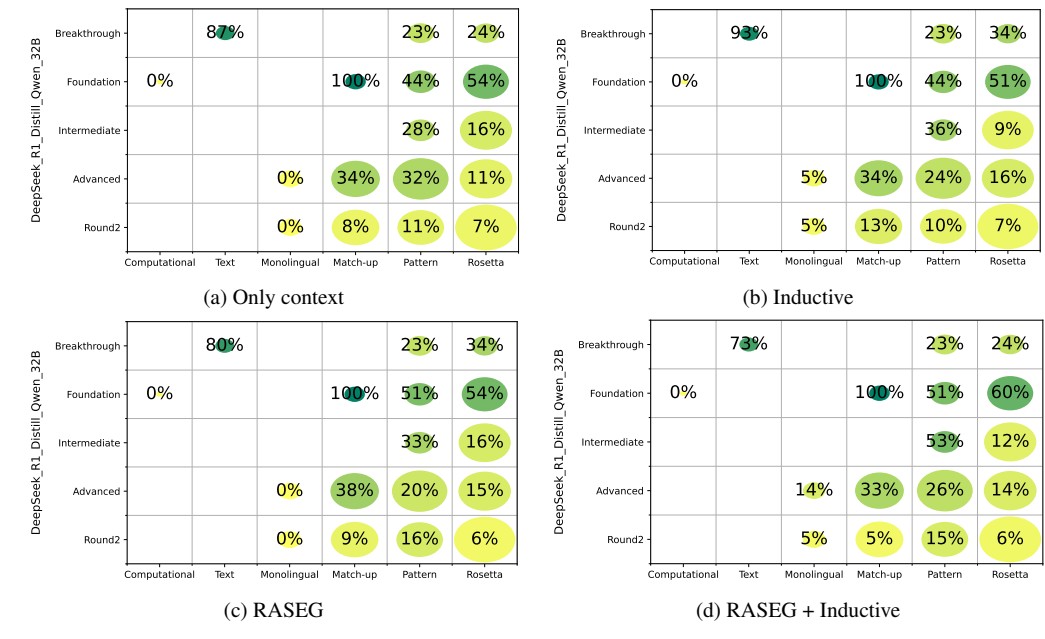

Figure 5: EM scores of various methods on different question types and difficulty levels of DeepSeek-R1-Distill-Qwen-32B.

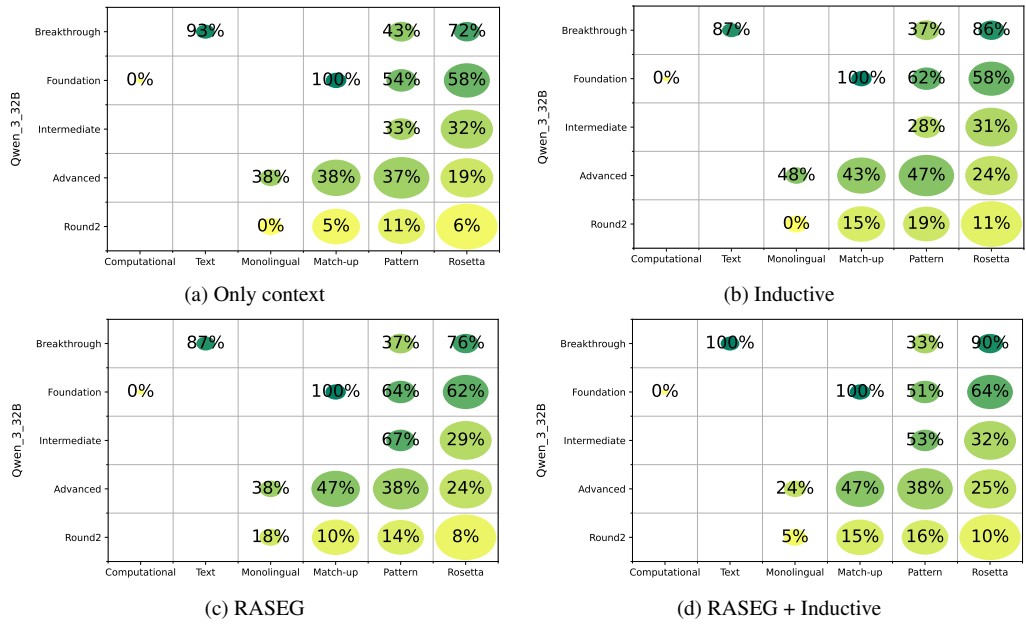

Figure 6: EM scores of various methods on different question types and difficulty levels of Qwen-3-32B.

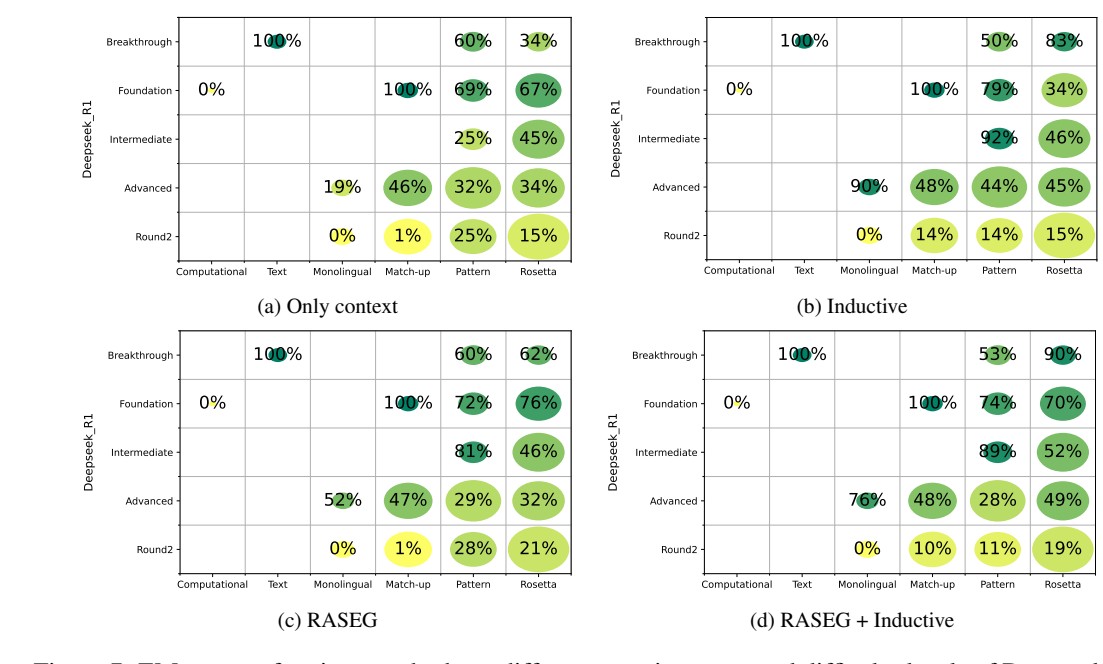

Figure 7: EM scores of various methods on different question types and difficulty levels of Deepseek-R1.

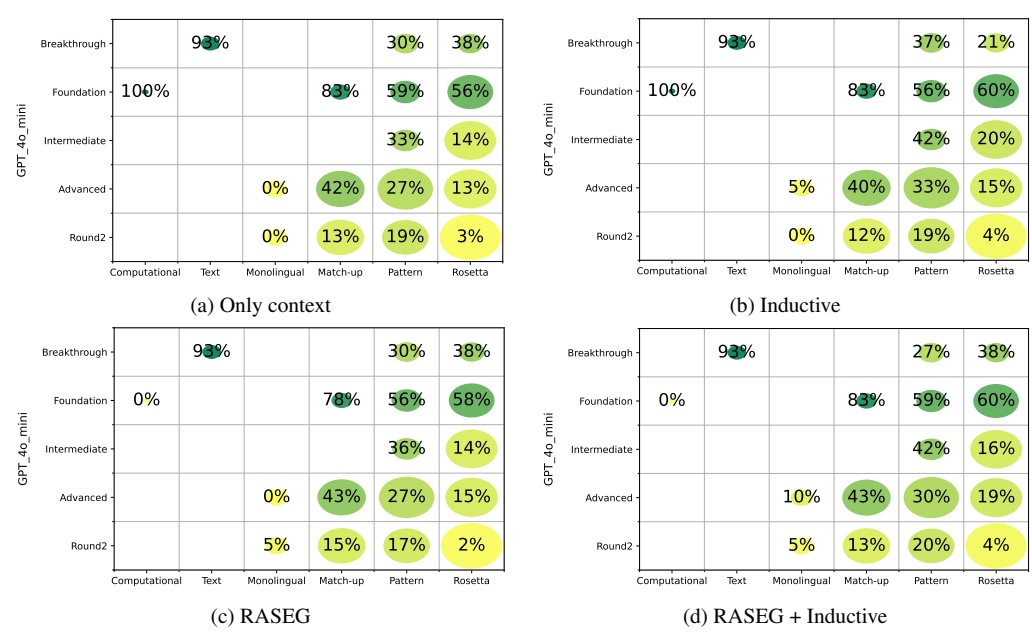

Figure 8: EM scores of various methods on different question types and difficulty levels of GPT-4o-mini.

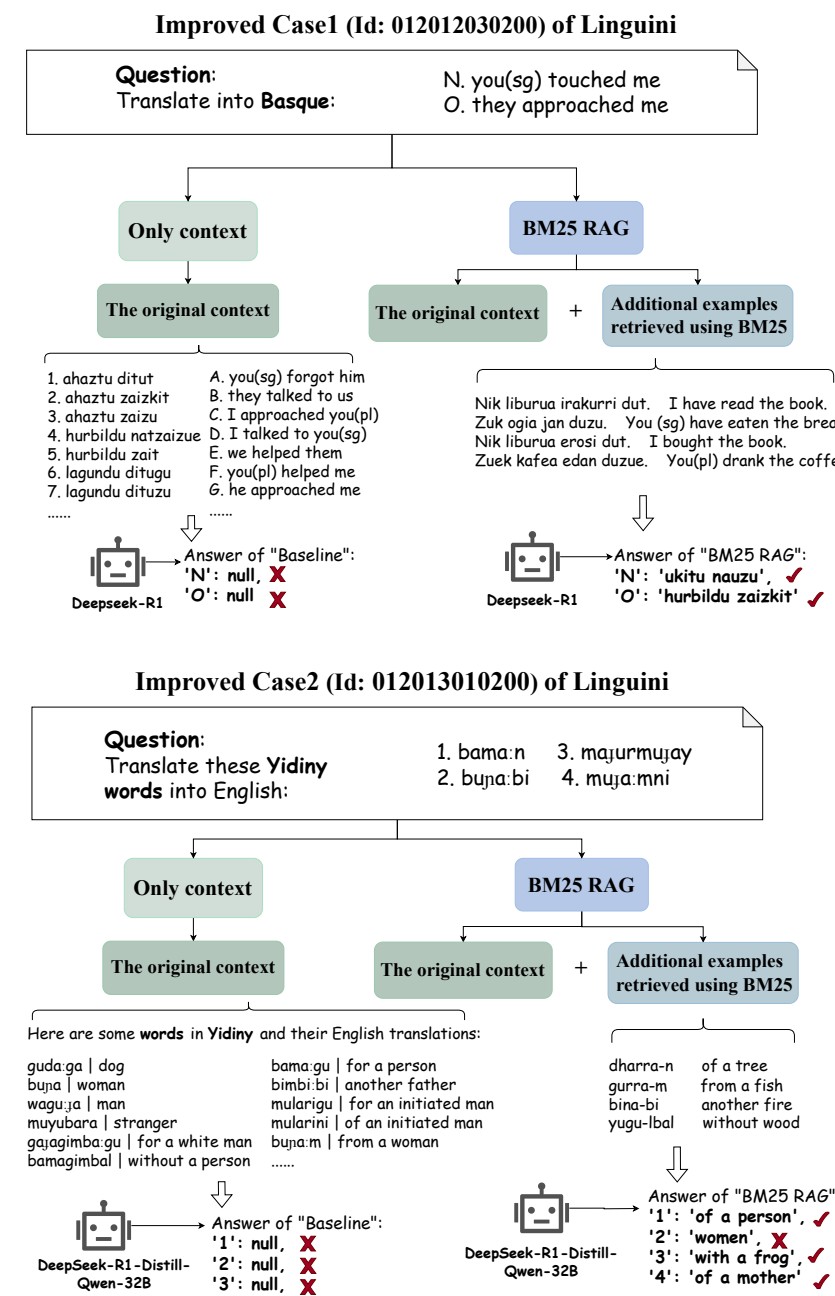

Figure 9: Examples of the only context and our framework on the Linguini in the sentence translation and words translation.

