# OpenReview forum: "Retrieval-Augmented Reasoning for Extremely Low-Resource Language Decipherment"
_ICLR.cc/2026/Conference — ICLR 2026 Conference Withdrawn Submission_

### Official Review · Reviewer_oPKA · 2025-10-20

**Soundness:** 3
**Presentation:** 3
**Contribution:** 2
**Rating:** 6
**Confidence:** 3

**Summary:**

This paper introduces a framework to tackle extremely low-resource language decipherment. The proposed method, Retrieval-Augmented Synthetic Exemplar Generation, addresses the limitations of existing in-context learning approaches by combining dynamic knowledge construction with task-aware retrieval augmentation. The framework uses LLMs to generate a diverse set of synthetic, task-specific examples of linguistic rules for a given low-resource language. Then, a semantic retrieval mechanism selects the most relevant examples for each test query to augment the context. This approach shifts the focus from learning language distributions to dynamically discovering and applying linguistic rules. The authors evaluate their framework on the LINGOLY and Linguini benchmarks, demonstrating competitive performance across various LLMs and outperforming existing baselines.

**Strengths:**

* The two-stage approach of "Dynamic Knowledge Construction" followed by "Task-Aware Retrieval Augmentation" is a natural solution to the problem of knowledge scarcity in extremely low-resource settings. It mimics the human process of hypothesis generation and analogical reasoning in decipherment tasks.

* The paper presents strong experimental results on two challenging benchmarks, LINGOLY and Linguini. The proposed RASEG method, especially when combined with inductive reasoning, consistently outperforms baseline models across various LLMs, including both open-source and proprietary models.

* The authors provide a detailed analysis of their results. This includes a comparison of different retriever types (BM25 vs. BGE), an ablation study on the number of retrieved examples, and a case study that shows the framework's advantages over baseline methods. The finding that the sparse retriever BM25 outperforms the dense retriever BGE is well-justified by the nature of the decipherment task, which benefits from exact lexical overlap.

**Weaknesses:**

* The "Dynamic Knowledge Construction" phase relies on a powerful LLM (Deepseek-R1) to generate high-quality synthetic exemplars. The performance of the entire framework is therefore heavily dependent on the capabilities of this generator model. The paper may need a discussion on how the quality of the generated knowledge base affects downstream performance.

* Although the offline generation of the knowledge base is a one-time cost per language, the process of generating a large and diverse set of exemplars could be computationally expensive.

**Questions:**

* The case study on the LINGOLY dataset for the Waanyi language is very insightful. Could you provide a similar detailed case study for the Linguini dataset?

* The paper mentions that the superiority of the BM25 retriever is due to its reliance on lexical overlap. Have you explored hybrid retrieval methods that combine the strengths of both sparse and dense retrievers?

* In the "Dynamic Knowledge Construction" phase, you employ a two-step quality control pipeline involving automatic filtering and rule consistency checks. Could you elaborate on the specifics of these checks?

* The paper says that the framework is "applicable to different types of Rosetta Stone problems". Have you considered evaluating your approach on historical decipherment tasks or other linguistic puzzles beyond the LINGOLY and Linguini benchmarks?

* How sensitive is the performance of the framework to the initial context provided in the problem?

---

### Official Review · Reviewer_R6g9 · 2025-10-31

**Soundness:** 2
**Presentation:** 2
**Contribution:** 2
**Rating:** 2
**Confidence:** 3

**Summary:**

The paper presents a framework for reasoning in extremely low-resource languages by combining synthetic data generation and retrieval-based augmentation. It first uses large language models to create diverse examples that simulate linguistic rules, then retrieves the most relevant ones for each query to guide reasoning. Experiments on LINGOLY and Linguini show that this method improves performance over previous baselines, even without in-language data. The authors find that sparse retrieval methods like BM25 work better than dense ones for precise linguistic reasoning, and conclude that generating and retrieving rule-based examples helps LLMs perform more human-like language understanding in unseen domains.

**Strengths:**

1. The work introduces a creative framework for extremely low-resource language reasoning by combining dynamic synthetic data generation with retrieval-based augmentation, offering a fresh perspective beyond traditional methods.
2. The paper is clearly written, with a well-structured explanation of each module and strong motivation for why retrieval and example generation complement each other.
3. The proposed framework offers significant potential for advancing language models' capabilities in extremely low-resource settings, where prior models often fail. It has the potential to enhance language understanding and application across novel linguistic domains, pushing the boundary of current AI research in natural language processing.

**Weaknesses:**

1. The work only involves models with 32B parameters or larger in the experiments, and the performance of the method on smaller models has not been demonstrated. This does not fully showcase the generalizability of the method.
2. The baselines used are relatively few. Relevant inductive reasoning methods should be introduced for a more comprehensive comparison.
3. In Section 3.1.1, how is the quality of the generated examples ensured? Although the work describes quality control methods in Section 3.2, there are no related experiments to support this.
4. The work seems to merely introduce the RAG technique into this new task, and the use of large models to generate knowledge for enhancement has already been widely applied in previous research. Therefore, the work lacks in demonstrating the novelty of the method. Could you further elaborate on the innovative aspects of this work?

**Questions:**

See Weaknesses

---

### Official Review · Reviewer_qbhS · 2025-11-01

**Soundness:** 2
**Presentation:** 1
**Contribution:** 2
**Rating:** 2
**Confidence:** 4

**Summary:**

The paper proposes RASEG, a framework for reasoning over extremely low-resource languages. It first uses DeepSeek-R1 to identify each language and its family, then generates synthetic exemplars at the sentence, phrase, and number levels to form a retrieval index. During inference, the model retrieves the top-k most relevant exemplars and combines them with the original context to perform analogical reasoning. Experiments on the LINGOLY and Linguini benchmarks show that this method achieves consistent improvements over few-shot and inductive baselines.

**Strengths:**

1. The paper addresses a novel and challenging problem in the decipherment of extremely low-resource languages, expanding retrieval-augmented reasoning to an unexplored and meaningful application domain with potential relevance to linguistic reasoning and endangered language preservation.

**Weaknesses:**

1. The overall writing quality of the paper is below the expected standard. Sections 3.1.1 and 3.1.2 merely provide procedural descriptions without sufficient analytical insight. The presentation lacks clarity and academic rigor required for publication. In addition, the paper appears to have used an incorrect or inconsistent formatting template.

2. The use of the term ISO 639-3 code may be unclear to readers without a background in linguistics.

3. The paper claims that the proposed task requires models to reason “without any prior knowledge” of the target language, relying solely on the limited contextual examples. However, in Section 3.1.1, the authors state that DeepSeek-R1 is used to identify the ISO 639-3 code of the low-resource language. This step implicitly relies on pretrained linguistic knowledge—the model must already recognize language features to output a valid ISO 639-3 label.

4. The conclusion that sparse retrieval (BM25) outperforms dense retrieval is not fully convincing, as the comparison involves only a single dense retriever (BGE). The authors should evaluate additional dense models to strengthen the generality and robustness of this claim.

5. The reported performance improvement appears to be relatively marginal. More inductive reasoning methods should be included for a more comprehensive and convincing comparison.

**Questions:**

See above.

---

### Official Review · Reviewer_wxZw · 2025-11-03

**Soundness:** 1
**Presentation:** 1
**Contribution:** 1
**Rating:** 2
**Confidence:** 3

**Summary:**

The paper proposes to use retrieval method for low-resource language decipher. Basically, the author propose to use a method that first generate examples to instantiate potential linguistic rules; then use retrieval method to retrieve the most relevant examples for each query to answer the question. The paper then tests on LINGOLY and Linguini benchmark, showing that this method can outperform existing baselines and advance low-resource language reasoning.

**Strengths:**

- Addresses an interesting task of low-resource language decipherment.

**Weaknesses:**

- The paper does not have enough novelty, it is basically just applying the existing classical "generating examples -> retrieve from examples pipeline" on the low-resource decipher task and reporting the result. This method is not new and applying this simple pipeline to a new task is not original enough.
- The paper does not compare with enough baselines from the RAG framework. Also should compare with simple reasoning methods to prove the effectiveness of their method.
- The paper is also not well written, eg in line 232 / 233 the left quote sign is wrong.

**Questions:**

- How does this pipeline compare with the state-of-the-art reasoning methods?

---

### Note · Authors · 2025-11-12

**Comment:**

Thank you very much to all the reviewers for taking the time to review. Through the review comments, we have identified the loopholes in our current work and would like to withdraw the manuscript for further improvement.

**Withdrawal Confirmation:**

I have read and agree with the venue's withdrawal policy on behalf of myself and my co-authors.